# See and Think: Disentangling Semantic Scene Completion

**Shice Liu**[1,2]    **Yu Hu**[1,2]    **Yiming Zeng**[1,2]    **Qiankun Tang**[1,2]    **Beibei Jin**[1,2]
**Yinhe Han**[1,2]    **Xiaowei Li**[1,2]

[1]State Key Laboratory of Computer Architecture
Institute of Computing Technology, Chinese Academy of Sciences
[2] University of Chinese Academy of Sciences
`{liushice,huyu,zengyiming,tangqiankun,jinbeibei,yinhes,lxw}@ict.ac.cn`

## Abstract

Semantic scene completion predicts volumetric occupancy and object category of a 3D scene, which helps intelligent agents to understand and interact with the surroundings. In this work, we propose a disentangled framework, sequentially carrying out 2D semantic segmentation, 2D-3D reprojection and 3D semantic scene completion. This three-stage framework has three advantages: (1) explicit semantic segmentation significantly boosts performance; (2) flexible fusion ways of sensor data bring good extensibility; (3) progress in any subtask will promote the holistic performance. Experimental results show that regardless of inputing a single depth or RGB-D, our framework can generate high-quality semantic scene completion, and outperforms state-of-the-art approaches on both synthetic and real datasets.

## 1 Introduction

Humans can understand unfamiliar circumstances quickly. For the scene shown in Figure 1, one can recover the shape of the sofa though some parts of it are occluded by clothes. To endow agents with this basic but important capability, semantic scene completion [1] is put forward, which predicts volumetric occupancy and object category of a 3D scene.

Early works on semantic scene completion is limited in visible surface partition [2, 3] and shape recovery without considering object category or environment context [4, 5]. Recently, [1, 6] agree scene completion and semantic labeling are tightly intertwined and simultaneously generate shapes and their categories. However, they not only exploit few semantic features, but are highly customized for a specific sensor. For a more effective and general framework, we present a novel modeling by borrowing ideas from human perception. The Gestalt psychologists proposed that segmentation, shape assignment and recognition were ordered serially and hierarchically and **lower-level cues form the substrate for higher-level cues** [7, 8]. Inspired by that, we treat 2D semantic segmentation as lower-level cues and leverage it to assist higher-level 3D semantic scene completion.

We propose a CNN-based framework to disentangle semantic scene completion. It sequentially accomplishes two subtasks, i.e., 2D semantic segmentation and 3D semantic scene completion. These two subtasks are connected by a 2D-3D reprojection layer. Considering that semantic segmentation is to acquire lower-level information as seeing and semantic scene completion is a higher-level task as thinking, we name it **See And Think Network (SATNet)**. Figure 1 shows its three modules.

Based on the framework, we provide various implementations. Note that all these implementations need a depth image, for it is required by the 2D-3D reprojection layer. For a single depth or RGB-D input, SATNet can finish semantic scene completion efficiently in a single-branch manner. Moreover, for the RGB-D input, two more effective double-branch implementations will be introduced.

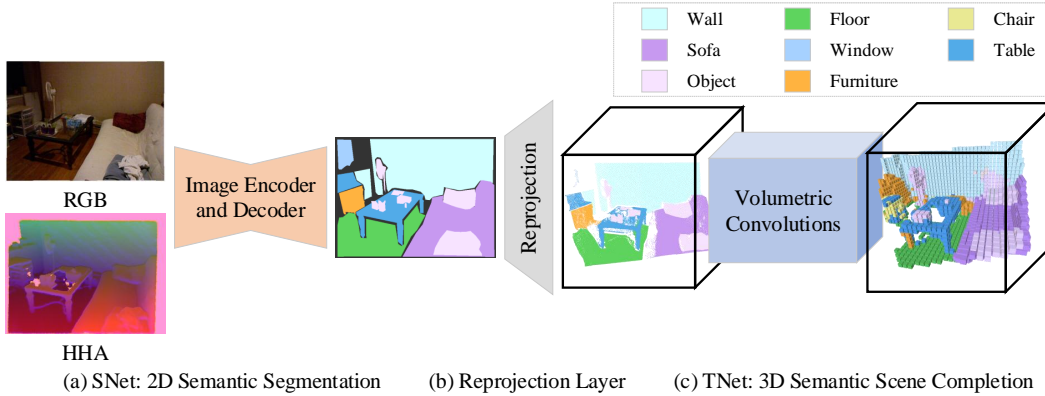

| | | | | | |
|---|---|---|---|---|---|
| ☐ | Wall | ☐ | Floor | ☐ | Chair |
| ☐ | Sofa | ☐ | Window | ☐ | Table |
| ☐ | Object | ☐ | Furniture | | |

(a) SNet: 2D Semantic Segmentation     (b) Reprojection Layer     (c) TNet: 3D Semantic Scene Completion

Figure 1: **See And Think Network.** It consists of (a) SNet, (b) a reprojection layer, and (c) TNet, sequentially carrying out 2D semantic segmentation, 2D-3D reprojection and 3D semantic scene completion. Given an RGB-D **or** HHA (a kind of depth encoding [9]) image, it generates the semantic volume representation of the scene.

Whichever input is given, SATNet outperforms the alternative approaches on both synthetic and real datasets. The advantages of SATNet are summarized below.

**Effectiveness.** We demonstrate that explicit semantic segmentation in SATNet boosts semantic scene completion, as is similar to human perception. Compared with state-of-the-art works, SATNet yields higher performance by 7% to 18% on the synthetic dataset and 3% to 4% on the real dataset.

**Extensibility.** The disentangled framework allows us to flexibly fuse different sensors within SATNet. We can replicate subnetworks as multiple branches according to the number of and the type of sensors, and then concatenate the corresponding features in SNet or TNet by lazy fusion.

**Evolvability.** This disentangled framework also taps the potential of evolving SATNet. As the complex task is partitioned into two easier subtasks and a simple transformation, it will be confirmed that progress in any subtask can heighten the effect of the entire framework.

## 2 Related Work

We briefly review related works on semantic segmentation and volume reconstruction, and then detail semantic scene completion from two perspectives, model fitting based completion and voxel reasoning based completion.

**Semantic segmentation.** Semantic segmentation is to acquire pixel-wise class labeling for an image. Deep learning based semantic segmentation contains two major approaches, i.e., image based and volume based. The former leverages the dense pixels of images to reach high performance [10–13], while the latter attempts to tap the potential of geometric information [14–16]. However, these methods consider pixels and voxels separately, thus not combining the advantages of images and volumes.

**Volume reconstruction.** Volume reconstruction can be finished by 3D convolution. With the help of large-scale 3D model repository like ShapeNet [17], a large number of works have been developed for exploiting the object shape prior [18–23]. By using large-scale 3D scene repository like SUNCG [1], learning based scene volume reconstruction is gradually tapped [6, 24].

**Model fitting based completion.** One effective approach to shape completion and semantic labeling is to fit 3D mesh models according to the observed surfaces [25–27, 2, 3]. Evidently, it is mainly limited in the capacity of the 3D model library. A large library can provide numerous models, but at the penalty of long time retrieval. A small library is very efficient, but may produce bad matching. To address the issue, [28–30] simplify 3D model fitting to 3D bounding box representation. Nevertheless, these methods sacrifice the details of objects for the speed of reconstruction.

**Voxel reasoning based completion.** Another popular approach is to complete and label voxels directly in the voxel space. By extracting features and integrating context, [31–33] use physical

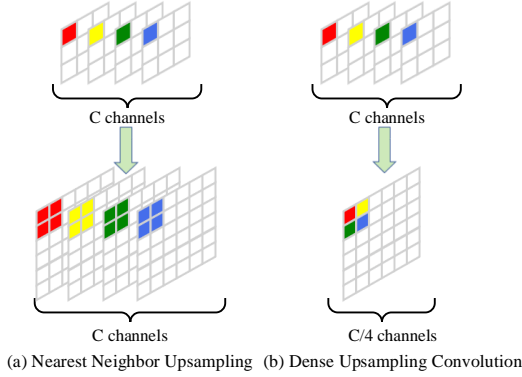
(a) Nearest Neighbor Upsampling (b) Dense Upsampling Convolution

Figure 2: **Comparison of two upsamplings.**
(a) Use the value of the nearest position in the corresponding feature map. (b) Shuffle four feature maps to a bigger one.

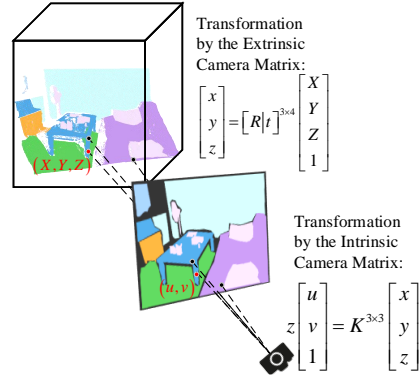

Figure 3: **2D-3D reprojection layer.** Given the intrinsic camera matrix and the depth image, we reproject the 2D semantic into 3D space and discretize it.

structure priors or conditional random field to reconstruct the unobserved part, but they are empirical or time-consuming. [34, 35] focus on multiview reconstruction and segmentation for outdoor scenes and [36, 37] manage to merge 2D semantic labels into 3D reconstructed surface via semantic visual SLAM. Obviously, multiview images are required in these approaches. Recently, [1] formulates the joint task of volumetric completion and semantic labeling as scene semantic completion, and proposes SSCNet to accomplish the task end-to-end. However, it takes a single depth image as input and does not take advantage of rich features of RGB. Afterwards, [6, 24] attempt to add RGB features into the network. Overall, in the aspect of task solving, these networks introduce a tight coupling between RGB and depth so that their extensibility and evolvability are constrained.

# 3 See And Think Network

SATNet is to attain higher-level semantic scene completion with the help of lower-level semantic segmentation. It contains three modules: **first**, a 2D semantic segmentation subnetwork which estimates the semantic by a single RGB or depth image (Sec. 3.1); **second**, a 2D-3D reprojection layer which transforms the 2D semantic into 3D space so as to complete the 3D scene (Sec. 3.2); **third**, a 3D semantic scene completion subnetwork that processes the voxels for semantic completion of the whole scene (Sec. 3.3). Finally, the double-branch RGB-D fusion in SNet or TNet is discussed (Sec. 3.4). Our source code is available at `https://github.com/ShiceLiu/SATNet`.

## 3.1 SNet: 2D Semantic Segmentation

The first issue we need to address is how to acquire semantic segmentation of the scene. Under the condition of the same memory consumption, images have higher resolution than volumes. In addition, 2D convolutions cost less time than 3D convolutions. Hense, we intend to tackle this subtask in 2D image space. The network of 2D semantic segmentation is named SNet.

For a better semantic segmentation result, we utilize an encoder-decoder architecture with skip connections to maintain the features under various receptive fields. The encoder is ResNet-101 [38], responsible for extracting multiscale features, and the decoder contains a series of dense upsampling convolutions [39]. As shown in Figure 2, the dense upsampling convolution outputs a feature map according to four feature maps of inputs, enabling details to be learned directly and avoiding the inaccuracy of interpolation.

The input of SNet is an $H \times W$ image $I$, which can be an RGB image $I_{rgb} \in \mathbb{R}^{3 \times H \times W}$ or a depth image $I_{depth} \in \mathbb{R}^{1 \times H \times W}$. Especially for the depth image input, it should be transformed to a three-channel HHA image $I_{depth}^* \in \mathbb{R}^{3 \times H \times W}$ mentioned in the method [9], for the purpose of keeping more effective information. SNet outputs $D$-channel semantic feature maps $SNet(I) \in \mathbb{R}^{D \times H \times W}$ which will be fed into the next module, the 2D-3D reprojection layer.

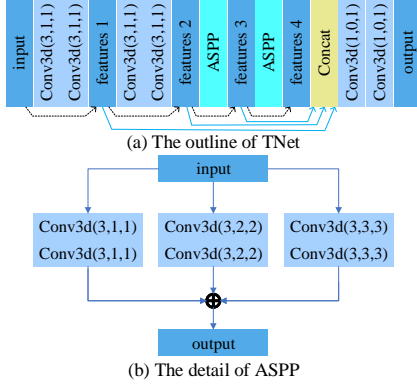

(a) The outline of TNet

(b) The detail of ASPP

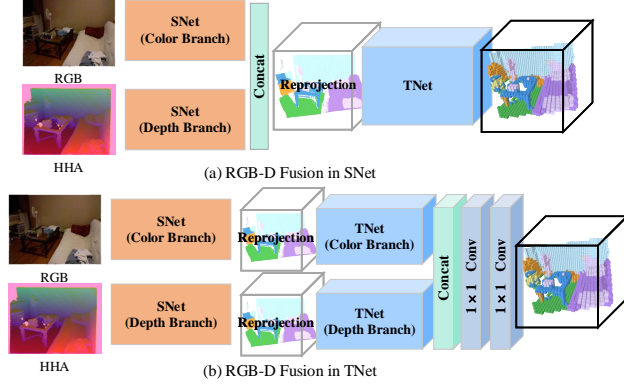

(a) RGB-D Fusion in SNet

(b) RGB-D Fusion in TNet

Figure 4: **The overall TNet.** The parameters of 3D convolution are kernel size, padding size and dilation size. The black dash lines denote adding while the blue solid lines denote concatenation.

Figure 5: **Two double-branch RGB-D fusion ways.** The double-branch fusion of RGB and depth for SATNet can be implemented in SNet or TNet: (a) SNetFuse is to fuse at the end of the SNet. (b) TNetFuse is to fuse at the end of the TNet.

### 3.2 2D-3D Reprojection Layer

Allowing for shape recovery in 3D space, the 2D-3D reprojection layer is a vital bond for mapping ample 2D semantic features to their corresponding 3D spatial positions. Provided the depth image $I_{depth}$, the intrinsic camera matrix $K \in \mathbb{R}^{3 \times 3}$ and the extrinsic camera matrix $[R|t] \in \mathbb{R}^{3 \times 4}$, each pixel $p_{u,v}$ (homogeneous coordinates are $[u, v, 1]^T$) in the image can be reprojected to an individual 3D point $p_{x,y,z}$ (homogeneous coordinates are $[X, Y, Z, 1]^T$) easily by camera projection equation $p_{u,v} = K [R|t] p_{x,y,z}$, as shown in Figure 3. After discretizing the 3D scene space into a volume whose size is $S_X \times S_Y \times S_Z$, we can establish the mapping $\mathbb{M}$ between 2D pixels and 3D voxels, and transform the image to the volume of scene surface.

In the period of forward propagation, the semantic feature vector $value_{u,v} \in \mathbb{R}^D$ for each pixel is assigned to its corresponding voxel via the mapping $\mathbb{M}$, while zero vectors are assigned to the empty foreground and the occluded areas in the scene. In this way, we obtain a volume $\pi (SNet (I)) \in \mathbb{R}^{D \times S_X \times S_Y \times S_Z}$, reflecting volumetric occupancy and semantic labeling of the 3D scene surface. In the period of backward propagation, only the gradients of voxels on the surface of the scene are propagated to $SNet (I)$, whereas the gradients of the voxels filled with zero vectors are not propagated. Overall, the propagation is so simple in either period that it can be regarded as reshaping feature maps. Accordingly, it takes little time to accomplish.

Furthermore, we rotate the scene volume to align with gravity based on the Manhattan assumption [40]. On one hand, it avoids tilted ground, walls or tables and provides better visual appearance. On the other hand, the shapes of objects will be simple due to not considering the angle of pitch or yaw. In practice, this rotation is implemented as an additional item in the extrinsic camera matrix.

### 3.3 TNet: 3D Semantic Scene Completion

Following the 2D-3D reprojection layer, TNet is proposed to accomplish higher-level semantic scene completion by the volume of lower-level semantic scene surface $\pi (SNet (I)) \in \mathbb{R}^{D \times S_X \times S_Y \times S_Z}$. As is shown in Figure 4, TNet completes a 3D scene step-by-step.

TNet is composed of two residual blocks [38], two atrous spatial pyramid poolings (ASPP) [13] and two $1 \times 1$ convolutions. The first four components compute semantic features of each voxel step-by-step. Then the four groups of feature maps are concatenated for the final category assignment by the two $1 \times 1$ convolutions. In the meantime, the residual block benefits quick convergence while ASPP aggregates multiscale features to enhance the capability of shape recovery. Thus, we successfully obtain the semantic completed scene volume $V = TNet (\pi (SNet (I))) \in \mathbb{R}^{C \times S_X \times S_Y \times S_Z}$, where $C$ denotes the number of categories we predict.

In SSCNet [1], the scene surface without semantic information is encoded into a volume for completion and all the convolutions have fixed receptive fields. Unlike SSCNet, TNet takes advantage of semantic scene surface and leverages multiscale feature extractors to yield better performance. Additionally, we will demonstrate these two measures are effective indeed.

### 3.4 Double-branch RGB-D Fusion

Not only can the three modules in SATNet cooperate with each other for accurate completion results, but also the modular framework of SATNet itself can make it easier to extend branches for fusing various sensors. As is shown in Figure 5, we introduce two double-branch fusion ways, SNetFuse and TNetFuse, to fuse RGB and depth.

**SNetFuse.** One double-branch fusion way is to fuse RGB and depth at the end of the SNet. The feature maps output by the color branch and by the depth branch are concatenated and then the concatenated feature maps will be reprojected to 3D space. The branches of RGB and depth share the same architecture but their parameters are not tied, enabling two branches to extract useful features for semantic segmentation respectively. Furthermore, it will be confirmed in the next section that better semantic segmentation is of great benefit to the final semantic scene completion.

**TNetFuse.** Another double-branch fusion way is to fuse RGB and depth at the end of the TNet, which integrates the two semantic scene completion results generated respectively by RGB and depth for a better one. Such a fusion manner can be regarded as boosting in the ensemble learning. Each of the two branches consists of SNet, 2D-3D reprojection layer, and TNet, but their parameters are not tied. Compared with SNetFuse, TNetFuse can achieve better completion results but at the expense of larger memory and time consumption.

### 3.5 Implementation Details

**Data Preprocessing.** Data preprocessing contains three parts. (1) Generation of volumetric labels: the size of the output volume is 4.8m (horizontally) $\times$2.88m (vertically) $\times$4.8m (in depth) and the volume is discretized with grid size 0.08m, resulting in a $60 \times 36 \times 60$ volume. The generation of scene volumetric labels is time consuming, so we prepare all the data offline. (2) Data balance: as the number of non-occupied voxels is much more than the occupied voxels, we randomly sample $N$ occupied voxels and $2N$ non-occupied voxels for training. (3) 2D-3D reprojection mapping calculation: we calculate the mapping $\mathbb{M}$ between 2D pixels and 3D voxels in advance.

**Training Paradigm.** We implement our framework in PyTorch. The training procedure consists of two steps. We first train 2D semantic segmentation with supervision. Then we initialize the weights of SNet and train SATNet end-to-end. We use cross entropy loss and SGD to optimize with a momentum of 0.9, a weight decay of 0.0001 and a batch size of 1. In addition, the learning rate of SNet and TNet is 0.001 and 0.01, respectively. It takes us around a week to accomplish the training period on GeForce GTX 1080Ti GPU.

## 4 Evaluation

### 4.1 Evaluation Basis

#### 4.1.1 Datasets

We evaluate our framework on two benchmark datasets, including the popular NYUv2 dataset [41] and the large-scale 3D scene repository SUNCG dataset [1].

**NYUv2.** The NYUv2 dataset, a real dataset, is composed of 1449 RGB-D images and is standardly partitioned into 795 training samples and 654 testing samples, each associated with an RGB and depth image. Due to no ground truth of volumetric occupancy and semantic labels in NYUv2 dataset, we follow [42] and [43] to attain the ground truth. NYUv2 is a challenging dataset, for it unavoidably has measurement errors and unmeasured areas in the depth images collected from Kinect. Inspired by [1], we pretrain the network on SUNCG before finetuning it on NYUv2 for better performance.

**SUNCG-D and SUNCG-RGBD.** The SUNCG dataset, a synthetic dataset, is composed of 45622 indoor scenes and one can acquire RGB-D images and semantic scene volumes by setting different

Table 1: Semantic scene completion results on NYUv2 dataset.

| method | Scene completion | | | Semantic scene completion | | | | | | | | | | | |
|---|---|---|---|---|---|---|---|---|---|---|---|---|---|---|---|
| | prec. | recall | IoU | ceil. | floor | wall | win. | chair | bed | sofa | table | tvs. | furn. | objs. | avg. |
| Lin [29] | 58.5 | 49.9 | 36.4 | 0.0 | 11.7 | 13.3 | 14.1 | 9.4 | 29.0 | 24.0 | 6.0 | 7.0 | 16.2 | 1.1 | 12.0 |
| Geiger [26] | 65.7 | 58.0 | 44.4 | 10.2 | 62.5 | 19.1 | 5.8 | 8.5 | 40.6 | 27.7 | 7.0 | 6.0 | 22.6 | 5.9 | 19.6 |
| Song [1] | 59.3 | **92.9** | 56.6 | 15.1 | **94.6** | 24.7 | 10.8 | 17.3 | 53.2 | 45.9 | 15.9 | 13.9 | 31.1 | 12.6 | 30.5 |
| Guedes [6]1 | 62.5 | 82.3 | 54.3 | - | - | - | - | - | - | - | - | - | - | - | 27.5 |
| Garbade [24] | **69.5** | 82.7 | **60.7** | 12.9 | 92.5 | 25.3 | 20.1 | 16.1 | 56.3 | 43.4 | 17.2 | 10.4 | 33.0 | 14.3 | 31.0 |
| Depth | 66.8 | 86.6 | 60.6 | 20.6 | 91.3 | 27.0 | 9.2 | **19.5** | 56.9 | **54.7** | 16.9 | 15.2 | 37.1 | 15.7 | 33.1 |
| RGBD | 69.2 | 81.2 | 59.5 | **22.5** | 87.0 | **30.0** | **21.1** | 17.9 | 52.4 | 44.5 | 15.1 | **19.5** | 36.0 | 17.3 | 33.0 |
| SNetFuse | 67.6 | 85.9 | **60.7** | 22.2 | 91.0 | 28.6 | 18.2 | 19.2 | 56.2 | 51.2 | 16.2 | 12.2 | 37.0 | 17.4 | 33.6 |
| TNetFuse | 67.3 | 85.8 | 60.6 | 17.3 | 92.1 | 28.0 | 16.6 | 19.3 | **57.5** | 53.8 | **17.7** | 18.5 | **38.4** | **18.9** | **34.4** |
| NoSNet | 67.3 | 84.4 | 59.5 | 19.8 | 94.5 | 28.1 | 0.7 | 15.4 | 50.8 | 43.2 | 15.7 | 11.0 | 31.9 | 7.7 | 29.0 |
| GTSNet | 64.4 | 92.5 | 61.3 | 22.3 | 94.5 | 32.0 | 27.6 | 22.7 | 58.4 | 58.9 | 24.8 | 28.0 | 49.4 | 29.8 | 40.8 |

1 Guedes et al. [6] did not provide class-wise results.

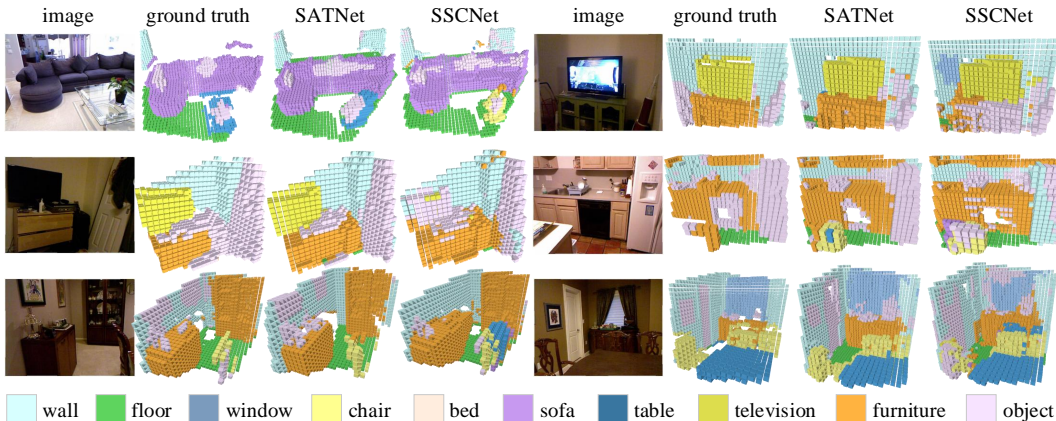

Figure 6: **Qualitative Results on NYUv2 Dataset.** We show the qualitative results generated by double-branch TNetFuse on NYUv2 Dataset. Compared with the results of SSCNet [1], SATNet gives more accurate semantic scene completions such as the windows and pictures on the wall.

camera positions. Song et al. [1] has provided a subset of SUNCG, made up of depth images and their volumes, so we call it SUNCG-D. Besides, we generate another subset SUNCG-RGBD, made up of RGB-D images and their volumes. SUNCG-D consists of 139368 training samples and 470 testing samples, while SUNCG-RGBD consists of 13011 training samples and 499 testing samples.

Although the SUNCG dataset is synthetic, the evaluation on the SUNCG dataset is quite significant. Considering that the limited number of models in the NYUv2 dataset might result in memorizing models, experiments in the large-scale 3D scene repository, *i.e.* SUNCG-D and SUNCG-RGBD, could relieve this issue to some degree.

#### 4.1.2 Metric

We mainly evaluate our framework with intersection over union (IoU) between our predictions and ground truth, and two tasks are considered: scene completion and semantic scene completion. For the former, we treat all voxels as binary predictions, i.e., occupied or non-occupied. For the latter, we pay attention to IoUs of each class and average them to get the mean IoU of semantic scene completion.

### 4.2 Experimental Results

#### 4.2.1 Comparison to Alternative Approaches

**Comparison on NYUv2 dataset.** Table 1 presents the semantic scene completion results on NYUv2 dataset with comparison to some alternative approaches. Lin et al. [29] uses 3D bounding boxes to approximate objects, while Geiger et al. [26] retrieves 3D models to find the best matching to represent the object. Song et al. [1] proposes SSCNet to predict volumetric occupancy and class labels simultaneously by a single depth image. Guedes et al. [6] utilizes two SSCNets for RGB and

Table 2: Semantic scene completion results on SUNCG-D dataset.

| method | Scene completion | | | Semantic scene completion | | | | | | | | | | | |
|---|---|---|---|---|---|---|---|---|---|---|---|---|---|---|---|
| | prec. | recall | IoU | ceil. | floor | wall | win. | chair | bed | sofa | table | tvs. | furn. | objs. | avg. |
| Song [1] | 76.3 | 95.2 | 73.5 | 96.3 | **84.9** | 56.8 | 28.2 | 21.3 | 56.0 | 52.7 | 33.7 | 10.9 | 44.3 | 25.4 | 46.4 |
| Depth | **80.7** | **96.5** | **78.5** | **97.9** | 82.5 | **57.7** | **58.5** | **45.1** | **78.4** | **72.3** | **47.3** | **45.7** | **67.1** | **55.2** | **64.3** |
| Depth (w/o i) | 77.7 | 95.1 | 74.9 | 97.2 | 80.9 | 52.7 | 44.4 | 33.6 | 69.6 | 62.5 | 34.0 | 25.5 | 49.0 | 39.3 | 53.5 |
| Depth (w/o a) | 80.9 | 95.3 | 78.0 | 97.8 | 82.1 | 56.8 | 57.4 | 41.1 | 72.6 | 71.3 | 45.7 | 44.0 | 60.3 | 51.9 | 61.9 |
| NoSNet | 79.2 | 94.5 | 75.8 | 97.5 | 80.8 | 56.7 | 29.4 | 33.0 | 61.5 | 65.0 | 34.4 | 28.8 | 50.2 | 38.0 | 52.3 |

Table 3: Semantic scene completion results on SUNCG-RGBD dataset.

| method | Scene completion | | | Semantic scene completion | | | | | | | | | | | |
|---|---|---|---|---|---|---|---|---|---|---|---|---|---|---|---|
| | prec. | recall | IoU | ceil. | floor | wall | win. | chair | bed | sofa | table | tvs. | furn. | objs. | avg. |
| Song [1] | 43.5 | 90.7 | 41.5 | 64.9 | 60.1 | **57.6** | 25.2 | 25.5 | 40.4 | 37.9 | 23.1 | 29.8 | 45.7 | 4.7 | 37.7 |
| Depth | 52.3 | 92.7 | 50.2 | 62.5 | 57.8 | 48.6 | **58.5** | 24.4 | 46.5 | 50.4 | 26.9 | **41.1** | 40.7 | 20.2 | 43.4 |
| RGBD | 49.8 | 94.3 | 48.3 | 59.0 | 45.0 | 46.0 | 50.6 | 24.9 | 42.0 | 49.0 | 26.8 | 40.8 | **46.6** | 22.4 | 41.2 |
| SNetFuse | **56.7** | 91.7 | **53.9** | **65.5** | **60.7** | 50.3 | 56.4 | 26.1 | **47.3** | 43.7 | **30.6** | 37.2 | 44.9 | **30.0** | 44.8 |
| TNetFuse | 53.9 | **95.2** | 52.6 | 60.6 | 57.3 | 53.2 | 52.7 | **27.4** | 46.8 | **53.3** | 28.6 | **41.1** | 44.1 | 29.0 | **44.9** |
| Depth (w/o i) | 49.2 | 93.8 | 47.6 | 45.4 | 57.3 | 48.0 | 38.4 | 20.3 | 36.9 | 36.6 | 18.2 | 26.7 | 36.6 | 15.2 | 34.5 |
| RGBD (w/o i) | 50.7 | 94.6 | 49.4 | 47.5 | 55.2 | 50.1 | 22.5 | 22.5 | 41.3 | 44.4 | 22.8 | 33.7 | 40.7 | 16.1 | 37.8 |
| Depth (w/o a) | 50.9 | 92.8 | 49.0 | 65.7 | 46.2 | 48.3 | 58.5 | 25.4 | 47.0 | 46.9 | 25.8 | 34.2 | 38.8 | 20.1 | 41.5 |
| RGBD (w/o a) | 48.3 | 94.8 | 47.1 | 64.1 | 51.0 | 45.1 | 48.8 | 23.7 | 41.1 | 42.5 | 20.1 | 41.7 | 34.4 | 24.3 | 39.7 |
| NoSNet | 50.8 | 92.8 | 48.8 | 66.6 | 56.7 | 41.7 | 30.6 | 22.7 | 47.1 | 36.4 | 22.1 | 25.2 | 30.7 | 13.4 | 35.7 |
| GTSNet | 53.4 | 95.0 | 52.0 | 63.7 | 55.6 | 48.0 | 61.8 | 33.9 | 51.3 | 57.4 | 38.0 | 43.6 | 48.5 | 28.8 | 48.2 |

depth respectively and combines the outputs of two SSCNet branches by concatenation. However, [6] directly processes low-resolution RGB scene surface in 3D space, resulting in losing lots of details. Garbade et al. [24] processes RGB by a series of 2D convolutions which is similar to our SNet, but directly follows SSCNet for depth. For SATNet, we experiment with four implementations: single-branch for depth input (Depth), single-branch for RGB-D input (RGBD), double-branch SNetFuse for RGB-D input (SNetFuse) and double-branch TNetFuse for RGB-D input (TNetFuse). Besides, we further do some ablation studies and discuss them in Sec. 4.2.2.

Compared to the other approaches in Table 1, the SATNet variants have the overall highest accuracy. For a single depth input, our approach produces more accurate predictions (Depth 33.1% vs. Song et al. [1] 30.5%). For RGB-D input, our approach gets even higher IoU by double-branch TNetFuse (TNetFuse 34.4% vs. Garbade et al. [24] 31.0%). In addition, for each category, we also acquire generally higher IoUs. Thus, 2D semantic segmentation for depth or RGB can promote the final completion. Figure 6 shows the qualitative results, where we complete six different scenes via double-branch TNetFuse and compare the results with SSCNet. Well as SSCNet works for many cases, it fails in the objects which are hard to distinguish from depth. For instance, SSCNet mistakes the television as another object in the second row of Figure 6, and fails to complete the pictures on the wall in the third row. In contrast, SATNet leverages RGB to overcome this difficulty.

**Comparison on SUNCG dataset.** Table 2 and Table 3 presents the quantitative results on SUNCG-D and SUNCG-RGBD dataset respectively. We mainly compare our framework with the benchmark approach Song et al. [1], because other approaches do not provide source data on SUNCG or source code. It can be seen that for the depth input, SATNet yields higher IoU for both SUNCG-D and SUNCG-RGBD datasets by 17.9% and 5.7% respectively, and the effect will be more prominent for the RGB-D input. Besides, the less noisy measurement on SUNCG dataset accounts for more accurate scene completions and higher mean IoU of semantic scene completion on SUNCG dataset than on NYUv2 dataset. We further do a lot of extra ablation studies which are detailed in Sec. 4.2.2.

### 4.2.2 Ablation Study

**Is disentangling semantic segmentation helpful?** [1] has proved that the semantic will help scene completion. Moreover, we intend to demonstrate that disentangling semantic segmentation is much more helpful for semantic scene completion, by means of presenting several ablation studies. Firstly, to exclude the effect of different inputs, we compare with [1] for the single depth input and compare with [6, 24] for the RGB-D input. Whatever input is provided, SATNet has better performance on both NYUv2 and SUNCG dataset. Secondly, to exclude the effect of the network structure, we maintain the same structure with SATNet but do not initialize SNet (represented as (w/o i)) with the weights trained on the semantic segmentation task. We examine in two single-branch situations, where SNet

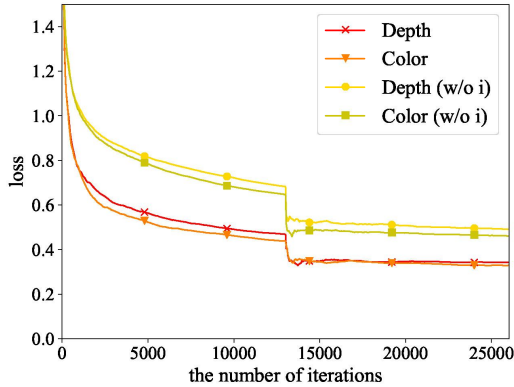

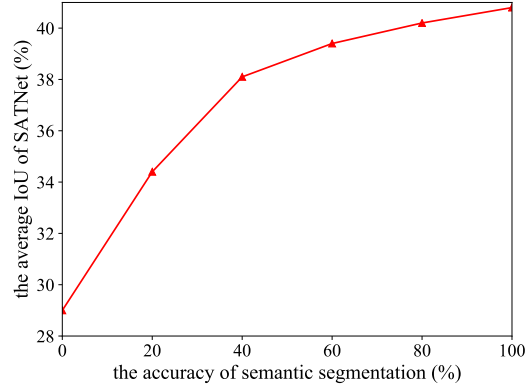

Figure 7: The convergence speed w/ or w/o semantic segmentation initialization.

Figure 8: The relationship of the average IoU and the segmentation accuracy.

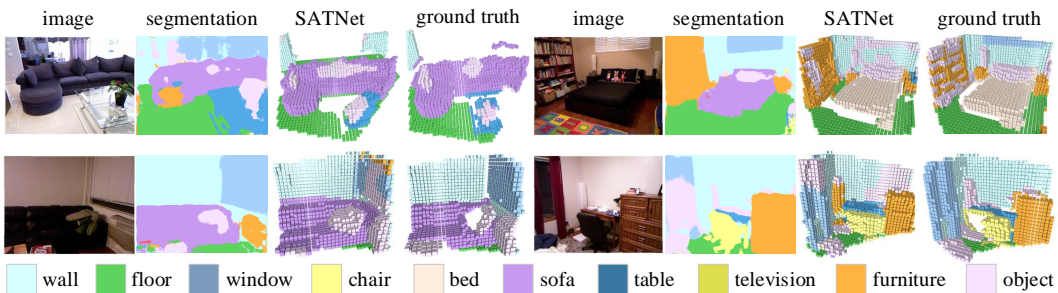

Figure 9: The corrections made by the TNet.

is provided with a depth image (Depth (w/o i)) or an RGB image (RGBD (w/o i)). As shown in Table 2 and Table 3, the network with initialization has higher performance than that without initialization. Thirdly, we find the explicit semantic segmentation could accelerate the speed of convergence. Figure 7 shows the loss of completion with 2D semantic segmentation initialization descends faster than that without initialization. Finally, to demonstrate disentangling semantic segmentation is helpful, Table 1 and Table 3 show the groundtruth semantic segmentation (GTSNet) could generate better completions than RGB or depth semantic segmentation, and much better than those without semantic segmentation (NoSNet).

**What is the effect of SNet?**   By reason that disentangling semantic segmentation is helpful, SNet plays an important role in generating semantic segmentation from various inputs. Moreover, in this part, we intend to probe to what extent the semantic segmentation benefits semantic scene completion. Figure 8 shows the relationship of the average IoU and the segmentation accuracy. While the accuracy of semantic segmentation is improved, the average IoU of semantic scene completion is bumped up. When the accuracy of segmentation reaches 60%, other parts instead of SNet would become the bottleneck and the average IoU of completion suffers slow growth. Whatever, better semantic segmentation always results in better semantic scene completion.

**What is the effect of TNet?**   Fed with semantic scene surfaces, TNet generates semantic scene completions. For the purpose of accurate completions, TNet is supposed to not only leverage semantic segmentation to assist completion, but also correct the mistakes made in the segmentation. Table 4 indicates the number and the ratio of correct/wrong completed voxels vary with correct/wrong segmented pixels (the pixels correspond to the voxels on the surface of the scene by the 2D-3D reprojection layer). As Table 4 shows, when the segmentation is correct, the probability of correct completion is 79%. When the segmentation is wrong, TNet is able to correct 55% mistakes occurred in the segmentation. To make the process of correction clear, Figure 9 shows some sensible corrections made by the TNet.

**Is aggregating multiscale features useful?**   As ASPP has attained great success in semantic segmentation, we intend to confirm that there is also its place in semantic scene completion. Hence,

|  | The result of semantic scene completion | |
| --- | --- | --- |
|  | Correct | Wrong |
| The result of semantic segmentation   Correct | 445514 (79%) | 119445 (21%) |
| The result of semantic segmentation   Wrong | 109656 (55%) | 89272 (45%) |

|  | SNetFuse | TNetFuse |
| --- | --- | --- |
| time for training (s/iter) | 0.9 | 1.3 |
| time for testing (s/iter) | 0.2 | 1.3 |
| memory for training (GB) | 7.3 | 13.7 |
| memory for testing (GB) | 2.2 | 3.5 |
| storage of parameters (GB) | 1.1 | 1.2 |

Table 4: The number and the ratio of correct/wrong completed voxels vary with the correct/wrong segmented pixels.

Table 5: Time and memory consumption for two double-branch fusions (SNetFuse and TNetFuse).

we compare the results with and without ASPP (represented as (w/o a)) in SATNet. Similarly, we examine on both depth input (Depth (w/o a)) and RGB-D input (RGBD (w/o a)). On SUNCG-D dataset, ASPP boosts IoUs in every class overall and yields 2.4% performance improvement averagely. On SUNCG-RGBD dataset, ASPP also brings 1.9% and 1.5% performance improvement for depth input and RGB-D input, respectively.

**Is there any difference for different inputs?** Although the mean IoU of single-branch RGB-D input and depth input on NYUv2 dataset are approximately equal, we find SATNet has different preferences in terms of two types of inputs. For the real scene dataset NYUv2, the IoU of each category differs. RGB-D inputs promote completing objects with unique colors and textures, such as windows and televisions; while depth inputs promote completing objects with clear shapes, such as beds and sofas. For the synthetic dataset SUNCG, due to the inauthentic RGB images, objects with large contrast will be overwhelmingly better, such as furniture and objects. Furthermore, the result of double-branch SNetFuse is better than single-branch RGB-D or depth input on both datasets, on account of more accurate semantic segmentation by RGB-D than by a single RGB or depth.

**Is there an absolutely best fusion way?** To answer this question, we take some extra metrics into account. Table 5 presents time and memory consumption for two fusions. They have approximately equal number of network parameters, but TNetFuse is much slower than SNetFuse. In fact, the time-consuming 3D convolutions account for the slow speed of TNet and numerous 3D feature maps account for its large memory consumption. Hence, we should choose the fusion approach that suits the actual demands, even though TNetFuse achieves higher performance indeed.

**Is it possible to evolve SATNet?** In order to demonstrate the evolvability of SATNet, we improve SATNet by individually enhancing the two subtasks. (1) Progress in SNet. Figure 8 shows what will happen if SNet produces more accurate segmentation. And the improvement in semantic segmentation will boost semantic scene completion to different extent. (2) Progress in TNet. In the previous discussion, we have shown that ASPP benefits semantic scene completion, which is an example that more effective components in TNet account for higher performance on completion.

**Limitations.** We do not discuss how the volumetric resolution influences semantic scene completion, owing to much larger memory requirement for large 3D volume. Intuitively speaking, low-resolution volumes might make some objects appear strange and restrict the effect. Moreover, to some extent, disentangling semantic segmentation from semantic scene completion is a recommendation given by us. It might be more effective and valuable to allow the network to spontaneously decide what subtasks it should disentangle.

# 5 Conclusion

In this paper, we propose SATNet, a disentangled semantic scene completion framework. It consists of SNet for 2D semantic segmentation, a 2D-3D reprojection layer, and TNet for semantic scene completion. We provide single-branch and double-branch implementations within the SATNet framework, which demonstrates the SATNet framework is effective, extensible and evolvable. Experimental results show SATNet outperforms the state-of-the-art approaches.

**Acknowledgments**

We thank Shuran Song for sharing the SSCNet results and the SUNCG development kits. This work is supported in part by National Natural Science Foundation of China under grant No. 61274030, and in part by Innovation Project of Institute of Computing Technology, Chinese Academy of Sciences under grant No. 20186090. Yu Hu is the corresponding author.

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
