[Reviews · NeurIPS 2018]

Reviewer 1



I want to like this paper. I think its well motivated and using a staged approach to this problem seems quite sensible. Additionally, they show quite good quantitative results on a challenging dataset. That being said, the model itself is not overly unique (not the worst thing) and the quantitative / qualitative results don't really cement the main purpose of the paper. Because the authors changed the network itself AND the approach, its not clear whether any superiority is due to multiple inputs (they use RGBD while other papers use RGB or D), a different network (perhaps this paper has a truly advantageous network structure) or the main idea: that predicting 2D first is better, is what really leads to good performance. === Main Ideas === The main idea of the work is to produce 3D semantic segmentation voxel grids, but do so via a staged approach: (1) 2D semantic segmentations are first estimated. (2) these are projected into 3D space (3) the unseen (occluded) voxels are inferred. The authors attempt three variants of this approach: 1. RGB or Depth --> Incomplete 3D --> Complete 3D 2. RGB and Depth pathways --> Incomplete 3D --> Complete 3D 3. RGB and Depth pathways --> Incomplete 3D from RGB, Incomplete 3D from depth --> Complete 3D === Quality === I like the idea overall: by breaking up the network into components, its much easier to debug and analyze than in pure end-to-end models. That being said, I really wish the authors had more clearly validated the efficacy of each model component. NYUDV2 is, in reality, a 2D dataset with depth. So if SATNet is good at predicting filled in voxel grids, it might do one of two things well: (1) estimate 2D semantic segmentations well or (2) somehow correct mistakes made in (1) but in 3D space. Filling in occluded pixels for this problem, while providing nice qualitative results, cannot help performance, because NYUDV2 has no labels for occluded pixels. Indeed, the clearest way to demonstrate that the proposed 3-stage approach is a good one would be to (a) show that better 2D segmentations result in better 3D results (surely this is true) or (b) that their 3D network is very good at fixing mistakes. If their model happens to do (a), then they need to clearly demonstrate this superiority in terms of 2D results and explain how they were able to obtain superior 2D results. If their model happens to do (b) this would make the paper ever more compelling. Overall, I'm left a bit wanting in terms of the quality. I want to like the paper and I think the authors approach is a potentially very good one, but I feel that much can be done to more clearly cement the main motivation of the work (staged approach is better than end-to-end) and demonstrate why indeed this would be the case. === Clarity === Overall, the paper is easy to read. Minor grammatical issues throughout but these do not impact the quality of the paper: - ln 77: "with the assist" => "with the assistance" - ln 81: "the semantic for semantic completion" => "the voxels for semantic completion" (I assume?) - ln 114: "it takes few time" => "it takes little time" There are a number of literary choices made by the authors that are extraneous and actually make the paper less clear overall: - ln 91: This is the first time I've seen the acronym DUC. Aside from paper [38], this is quite uncommon and actually makes the paper less clear. I would recommend just referring to the Decoder Network. - Naming / Branding the entire approach SATNet is fine. Giving names to the individual sub-components (SeeNet, ThinkNet, SeeNetFuse, ThinkNetFuse) seems extraneous and very confusing. Its far clearer to simply describe the components of the network, Figure 5 does a very good job of showing this off without the need to sub-brand parts of the network. - ln 85: "Images have higher resolution than volumes.". This is plainly untrue. Either one can be constructed to have an arbitrarily high resolution. It happens that in practice, researchers tend to use voxel grids that have a smaller spatial resolution, mostly due to computational constraints. This might be what the authors meant? Minor Nit: the following reference is missing: A contour completion model for augmenting surface reconstructions which does volume / voxel reconstruction, partially semantically. === Originality === The paper is mildly original. Many works solve problems in a staged approach. To my knowledge, this is the first to do so on this particular task. === Significance === The problem of 3D semantic scene completion is an increasingly popular one. The authors demonstrate some very promising quantitative results which would be generally interesting to the community.

Reviewer 2



Post rebuttal: I read the rebuttal and other reviews. I will keep the initial rating (6) since there are some issues that still need to be addressed: - Obtaining intrinsic and extrinsic parameters of the camera is not trivial since in many cases we do not have access to the camera to calibrate it. - There are several revisions suggested by the reviewers that need to be incorporated. ************************** Paper summary: The paper tackles the problem of semantic scene completion, where the idea is to find 3D regions corresponding to an object (occluded or not) and its category. The proposed model has three modules. First, it performs 2D semantic segmentation. Then, it maps the segmented image to the 3D space. Finally, it assigns category labels to the voxels of the scene. Paper strengths: - The proposed model outperforms the previous models. - The paper provides an extensive set of ablation experiments. Paper weaknesses: - My understanding is that R,t and K (the extrinsic and intrinsic parameters of the camera) are provided to the model at test time for the re-projection layer. Correct me in the rebuttal if I am wrong. If that is the case, the model will be very limited and it cannot be applied to general settings. If that is not the case and these parameters are learned, what is the loss function? - Another issue of the paper is that the disentangling is done manually. For example, the semantic segmentation network is the first module in the pipeline. Why is that? Why not something else? It would be interesting if the paper did not have this type of manual disentangling, and everything was learned. - "semantic" segmentation is not low-level since the categories are specified for each pixel so the statements about semantic segmentation being a low-level cue should be removed from the paper. - During evaluation at test time, how is the 3D alignment between the prediction and the groundtruth found? - Please comment on why the performance of GTSeeNet is lower than that of SeeNetFuse and ThinkNetFuse. The expectation is that groundtruth 2D segmentation should improve the results. - line 180: Why not using the same amount of samples for SUNCG-D and SUNCG-RGBD? - What does NoSeeNet mean? Does it mean D=1 in line 96? - I cannot parse lines 113-114. Please clarify.

Reviewer 3



This paper studies the task of semantic scene completion. Given a D / RGB-D image, the semantic scene completion problem is to be predict semantic labels for the full 3D volume of the scene. The current paper proposes a way to fuse information from discriminative 2D image-based CNNs with 3D information from the depth image to produce the semantic scene completion output. The paper evaluates the proposed approach on NYUD2, and SUNCG datasets and is able to outperform existing approaches for this task. The paper additionally also reports a number of ablations for this task. I generally like the paper. The central contribution is to augment the 3D reasoning with discriminative features computed from the 2D by projecting these features into the voxel map using the camera intrinsics and extrinsics. Kar et al [23] have proposed a very similar feature unprojection operation in context of reasoning from RGB images alone, but I find the empirical evaluation of this idea for the task of semantic understanding interesting. The paper shows large improvements over past works in varies contexts (D / RGB-D). The paper however has some shortcomings: Experimental evaluation on NYU Dataset. I will like to note that the evaluation for Semantic Scene Completion on the NYU dataset is somewhat flawed. The NYU dataset comes with approximate CAD models that are aligned to images. The same approximate models are used between the training and the testing sets, thus the problem as is framed on NYU is somewhat game-able by being treated as a retrieval problem. I worry that some of this might also happen with expressive CNN models that can memorize and regurgitate models from the training set. Past works such as [1,6,24] suffer from the same issue, but this issue should be noted in the paper, and these evaluations should be taken with a grain of salt. The same issue also plagues experiments on the SUNCG dataset (though possibly to a somewhat smaller extent given the relatively large number of CAD models in use there). Furthermore, I believe the paper should more properly attribute credit to [23]. The paper should also reconsider the choice of names for SeeNet and ThinkNet, I don't find them particularly evocative of what is happening in the paper.